# Surface Defect Detection of Aluminum Profiles Based on Multiscale and Self-Attention Mechanisms

**DOI:** 10.3390/s24092914

**Published:** 2024-05-02

**Authors:** Yichuan Shao, Shuo Fan, Qian Zhao, Le Zhang, Haijing Sun

**Affiliations:** 1School of Intelligent Science and Engineering, Shenyang University, Shenyang 110044, China; shaoyichuan@syu.edu.cn (Y.S.); snowise@syu.edu.cn (L.Z.); 2School of Information Engineering, Shenyang University, Shenyang 110044, China; fanshuo@syu.edu.cn; 3School of Science, Shenyang University of Technology, Shenyang 110044, China; qzhao@uow.edu.au

**Keywords:** aluminum profile defect detection, gaussian difference pyramid, self-attention mechanism, transfer learning, residual network, dilated convolution

## Abstract

To address the various challenges in aluminum surface defect detection, such as multiscale intricacies, sensitivity to lighting variations, occlusion, and noise, this study proposes the AluDef-ClassNet model. Firstly, a Gaussian difference pyramid is utilized to capture multiscale image features. Secondly, a self-attention mechanism is introduced to enhance feature representation. Additionally, an improved residual network structure incorporating dilated convolutions is adopted to increase the receptive field, thereby enhancing the network’s ability to learn from extensive information. A small-scale dataset of high-quality aluminum surface defect images is acquired using a CCD camera. To better tackle the challenges in surface defect detection, advanced deep learning techniques and data augmentation strategies are employed. To address the difficulty of data labeling, a transfer learning approach based on fine-tuning is utilized, leveraging prior knowledge to enhance the efficiency and accuracy of model training. In dataset testing, our model achieved a classification accuracy of 97.6%, demonstrating significant advantages over other classification models.

## 1. Introduction

The aluminum profile manufacturing industry is a cornerstone sector in the Nanhai district of Foshan, playing a critically important role in fostering the healthy development of the local economy. However, with the continual elevation of quality standards for aluminum profiles on the market, surface defects on aluminum profiles have emerged as a pressing challenge requiring urgent attention from manufacturers and regulatory authorities. The surface defects on aluminum profiles not only impact the aesthetic appeal of the products but also have the potential to lead to performance degradation and even pose safety hazards [1]. Therefore, quality control and defect detection are of paramount importance in ensuring product quality and market competitiveness.

Defect detection is a crucial step in ensuring the quality of industrial production [2]. In the evolution of surface defect detection technologies for metals, there has been a progression from manual inspection to automated detection based on traditional machine learning algorithms, and further to automated detection based on deep learning algorithms [3]. In many cases, defect detection on product surfaces heavily relies on manual visual inspection [4]. This method is time-consuming, labor-intensive, and susceptible to subjective interference. Specifically, the surfaces of aluminum profiles often exhibit textures similar to defects, making it challenging for the human eye to make accurate judgments. This not only results in lower detection efficiency but also higher rates of false positives and false negatives. Moreover, the high-speed production lines in large-scale industrial manufacturing further complicate defect detection, rendering this method increasingly inadequate [5].

Defect classification is a fundamental task in industrial inspection, aiming to identify the category of defect images [6]. With the advancement of computer vision, defect detection methods for aluminum profiles have primarily relied on traditional image processing techniques, such as edge detection [7], texture analysis [8], and shape recognition [9]. The development of automatic surface defect detection technology [10] has garnered attention from the academic community as image technology has progressed. In comparison to manual inspection, automated defect detection systems offer numerous advantages, including prolonged continuous operation, consistent detection results, and functionality in harsh environments involving high temperatures and dust [11]. While these methods can detect some common surface defects to a certain extent, their effectiveness is limited when it comes to detecting complex and variable defect types with irregular shapes. Defect detection based on traditional methods has inherent limitations; for instance, variations in image illumination, brightness, and quality can significantly impact the results [12].

In recent years, with the continuous development of deep learning technology, image classification methods based on deep learning have achieved notable success, exemplified by architectures such as AlexNet [13], VGGNet [14], ResNet [15], DenseNett [16], and others. Convolutional neural networks (CNNs) have demonstrated remarkable achievements in image recognition tasks, inspiring considerable exploration of their potential in defect detection. Numerous scholars have begun to investigate the application of deep learning to defect detection.

Shakeel et al. [17] introduced an adaptive multiscale attention module for effectively aligning feature maps. Zhang et al. [18] proposed MCNet, a model employing pyramid pooling to capture multiscale contextual information. Baffour et al. [19] introduced a self-attention module focusing on handling spatial positional information in feature maps. Yang et al. [20] proposed an end-to-end detection method for aluminum profile surface defects based on ResNet and attention mechanisms. Kou et al. [21] developed an end-to-end defect detection model based on YOLO-V3, incorporating an anchor-free feature selection mechanism to choose the most suitable feature scale for model training. They also introduced specially designed dense convolution blocks to enhance feature information and reduce computation time. In [22], an algorithm based on Faster R-CNN [23] was proposed, aiming to achieve the detection of surface defects on aluminum profiles at different scales.

Despite significant advancements being made in deep learning methods for surface defect detection on aluminum, several challenges persist. For instance, the high cost of data annotation poses a challenge, as deep learning models require a substantial amount of labeled training data [24]. Aluminum surface defects are predominantly small targets, with some being extremely diminutive and closely resembling the background color, making it challenging for detection models to extract key features [25]. The irregular and highly variable shapes of aluminum surface defects make it difficult to acquire fundamental information using traditional convolutional kernels. Disparities in the quantity of different types of defect images can introduce errors in the training set, subsequently interfering with model performance. Sparse defect samples in industrial settings make it challenging to adequately and effectively train the model, resulting in overfitting and poor generalization performance, ultimately impacting detection accuracy [26].

This study aims to leverage deep learning technology to extract multiscale features of surface defects on aluminum profiles using a Gaussian difference pyramid architecture. An adaptive learning module is incorporated to better capture crucial information in images, enhancing the perception of key areas and thereby improving the accuracy and generalization capability of aluminum profile defect detection. The ResNet50 model is employed in transfer learning, capitalizing on its pre-trained feature extraction capabilities on large-scale image data to expedite the training process on new tasks and enhance model generalization performance.

Our main contributions can be summarized as follows:(1)We made a small-scale dataset of surface defects on aluminum profiles using a CCD camera, which provides high-quality digital image data.(2)The model accepts input images through a Gaussian difference pyramid, generating multiple difference images at different scales. These difference images reflect details and features of the original image at various spatial scales. Through Gaussian blurring and differencing operations, the extraction and representation of the multiscale features of the image are facilitated.(3)The introduction of a self-attention mechanism enables the model to better comprehend crucial information in the images. By learning and applying attention weights, the model can emphasize key features, thereby enhancing detection performance.(4)The incorporation of a residual network with dilated convolutions, coupled with transfer learning based on fine-tuning, enhances neural network training efficiency, addresses the issue of vanishing gradients, and achieves the efficient classification of aluminum profile defect images in small-scale and imbalanced datasets.(5)Experimental validation on the task of surface defect detection in aluminum profiles demonstrates the outstanding performance of the model in terms of accuracy and efficiency. Relative to traditional methods and other deep learning models, it significantly improves classification accuracy, providing a more reliable solution for industrial quality control.

The structure of this paper is as follows: Section 1 introduces the theme of aluminum profile surface defect classification, outlines the current challenges, and proposes the AluDef-ClassNet model. Section 2 provides an overview of the relevant principles and methods of the model. Section 3 describes the process of data collection. Section 4 provides a detailed description of the experimental setup and results analysis. Finally, Section 5 concludes the paper.

## 2. Principles and Methods Related to the Study

Surface defect detection in aluminum profiles faces multiple challenges in practical applications. These challenges include variations in the sizes of different aluminum profiles and defect magnitudes, making it difficult for traditional neural networks to handle such multiscale scenarios. Additionally, diverse lighting conditions contribute to variations in the appearance of aluminum profile surfaces, rendering traditional models sensitive to lighting changes. Environmental noise and device-related noise during the image acquisition process can impact image quality, with traditional methods being susceptible to such influences. Therefore, advanced deep learning techniques are required for surface defect detection in aluminum profiles to address these complexities and diversities. To this end, we propose a network model called AluDef-ClassNet to improve detection accuracy. This model integrates Gaussian difference pyramids, self-attention mechanisms, and dilated convolutions to enhance the capture and representation of image features. This section will introduce the construction principles and methods of the AluDef-ClassNet model, as well as its application in aluminum extrusion defect detection.

### 2.1. Construction of Gaussian Difference Pyramid

The Gaussian difference pyramid, as an effective method, finds widespread applications in image processing, particularly demonstrating outstanding performance in capturing image features in a multiscale manner. When addressing real-world image problems, objects often appear at different scales, and the utilization of the Gaussian difference pyramid allows for a more comprehensive identification and analysis of these features [27]. By applying Gaussian blurring and downsampling operations to the image, it constructs an image pyramid at different scales, effectively capturing and representing information about the image across multiple scales. This proves instrumental in addressing practical problems involving scale variations, such as target detection and image segmentation. The multiscale feature extraction approach exhibits significant advantages in dealing with complex scenes and multiscale objects, providing a powerful tool for the successful implementation of image processing tasks.

The Gaussian difference pyramid is constructed by computing the difference between adjacent levels of images in the Gaussian pyramid, as illustrated in Figure 1. It is an image processing technique that initially involves the application of Gaussian blur filters at each level, creating a set of images with varying degrees of blurring [28]. These images constitute the Gaussian pyramid. Subsequently, the difference between adjacent scale-level images is computed to form difference images, containing detailed information about the image at different scales. These difference images collectively constitute the Gaussian difference pyramid.

By using the Gaussian difference pyramid to decompose the image, multiple images with different resolutions can be generated. The Gaussian difference pyramid comprises multiple pyramids, each containing several layers. Each pyramid is built on the foundation of a Gaussian pyramid, with its levels composed of a series of Gaussian pyramid layers [29]. The decomposition process of the Gaussian difference pyramid is outlined as follows:

Step 1: Initialize *a* = 0.

Step 2: Upsample the standard image A(*x*, *y*) to obtain the first layer image *g*_0,0_ of the Gaussian pyramid.

Step 3: Initialize *b* = 0 and *x* = 0.

Step 4: Convolve the Gaussian kernel *G_x_* with the image *g_a_*_,0_ [30]:(1)$$G_{x}\left(x,y,\sigma_{x}\right)=\frac{1}{2\pi \sigma_{x}^{2}}e^{\frac{{\left(x-x_{0}\right)}^{2}+{\left(y-y_{0}\right)}^{2}}{2\sigma_{x}^{2}}}$$
(2)$$g_{a,b+1}(x,y)=g_{a,b}(x,y)⊗G_{x}\left(x,y,\sigma_{x}\right)$$
where *σ_x_* is the smoothing parameter.

Step 5: Obtain the Gaussian difference image by subtracting Gaussian image *g_a_*_,*b*_(*x*, *y*) from Gaussian image *g_a_*_,*b*+1_(*x*, *y*) [31]:(3)$$d_{a,x}(x,y)=g_{a,b}(x,y)-g_{a,b+1}(x,y)$$

Step 6: *b* = *b* + 1, *x* = *x* + 1, repeat Steps 4 and 5 iteratively. When *b* > *n* − 1 and *x* > *n* − 2, proceed to Step 7.

Step 7: Downsample image *g_a_*_,0_ to obtain the Gaussian image of the *a* + 1 layer. If *a* = *a* + 1, return to Step 3. The decomposition process concludes when condition *a* > *m* − 1 is satisfied.

For the Gaussian difference pyramid, the parameters are set as follows: the input image size is 224 × 224, the number of Gaussian kernels is three, each Gaussian kernel size is 3 × 3, and the number of pyramid layers is three. At each level of the pyramid, the original image is first subjected to Gaussian blurring to reduce noise and details. Subsequently, the smoothed image is downsampled with a downsampling stride of two. For the *i*-th level of the pyramid, the output size can be calculated using the following Formula (4):(4)$$H_{i}=\frac{H}{s^{i}},W_{i}=\frac{W}{s^{i}}$$
where *H_i_* and *W_i_* are, respectively, the height and width of the *i*-th layer of the pyramid image, *s* is the downsampling stride, and *H* × *W* is the size of the input image. Thus, the resulting downsampling image sizes are 112 × 112, 56 × 56, and 28 × 28, respectively. Through the difference calculation, the local features of the image at different scales can be obtained, resulting in images of sizes 112 × 112 and 56 × 56. In the fusion module, the 56 × 56 images are enlarged to 112 × 112 using bilinear interpolation and then concatenated horizontally with the 112 × 112 images to obtain a 224 × 112 image. Subsequently, the obtained 224 × 112 image is concatenated vertically with another 224 × 112 image to obtain a 224 × 224 image. Finally, the difference images are pixel-wise weighted and summed with the original images to obtain the fused feature map.

### 2.2. Self-Attention Mechanism Module

The self-attention mechanism [32] is a critical technique in deep learning for handling image data. It allows the model to dynamically allocate attention weights to different positions during the learning process. By computing the relevance of each position in the input image with other positions, the self-attention mechanism enables the model to flexibly capture spatial relationships in the input image, enhancing the modeling capability for complex image features. This technique has been widely applied in areas such as image classification and object detection.

Here, initially, the feature map is fed into a small convolutional layer, resulting in a feature map with the same size as the original one but with more channels. Subsequently, a Sigmoid activation function is applied to map the values of each channel in this feature map to the range of 0 to 1, forming a weight map. This map informs the model about which channels or features are more important in subsequent computations. Finally, this weight map is applied to the original feature map by adjusting the weight of the features through the multiplication of each channel’s feature values with the corresponding attention weight at that position. The definition is provided by Formula (5).
(5)$$Z=Softmax(\frac{C_{M}\cdot W_{q}\cdot {\left(C_{M}\cdot W_{k}\right)}^{T}}{\sqrt{d_{k}}})\cdot (C_{M}\cdot W_{\nu })$$
where *C*_M_ is the input feature map, and *W_q_*, *W*_k_, and *W_v_* are the learnable weight matrices used for linearly transforming the input features to compute queries, keys, and values. $\sqrt{d_{k}}$ is the scaling factor, where *d*_k_ represents the dimension of the keys. The Softmax function normalizes the computed attention weights to ensure that the sum of the weights is one. “·” denotes matrix multiplication. The entire process is illustrated in Figure 2.

This is akin to allowing the model to automatically determine which features are more conducive to the current task, thereby enhancing the model’s performance, especially when dealing with complex tasks and images with diverse features. In our aluminum profile defect detection model, the self-attention mechanism aids in better understanding the correlation of different regions in the image, improving the accuracy of defect detection.

### 2.3. Residual Network Module

In the residual block, initially proposed by Kaiming He et al. [15], two types of residual blocks are usually included, as shown in Figure 3, which introduces the concept of skipping connections. The key idea of the residual block is to allow for the direct transmission of information between different layers in a neural network, rather than passing through multiple non-linear layers as in traditional neural networks. This means that during the backpropagation process, gradients can more easily propagate back through the network, alleviating the vanishing gradient problem and enabling the training of very deep networks. This innovation has had a significant impact on the field of deep learning, making it easier to train and optimize deeper and more complex neural network models, thereby improving the performance of tasks such as image recognition and other computer vision tasks.

#### 2.3.1. Transfer Learning Based on Fine-Tuning

In the field of deep learning, constructing a well-performing neural network model typically requires a significant amount of training data. However, not all tasks have access to a sufficient quantity of labeled data, and data acquisition often comes with high costs. To address the challenge of difficult data labeling, both semi-supervised learning [29,30] and transfer learning have been proposed as effective solutions. In transfer learning, this approach involves loading pre-trained model parameters from a similar task into a new task, achieving parameter initialization. This reduces the reliance on training data, allowing satisfactory model performance to be achieved with less data and training iterations. Common datasets used for pre-training models include ImageNet, which comprises over 1.2 million images. Various widely used models have been trained on this dataset, making it suitable for transfer learning and facilitating more efficient model training.

This study aims to classify images of defects in aluminum profiles. The data used are sourced from the actual production process of the aluminum profile manufacturing industry. However, the dataset is relatively small and exhibits class imbalance. Given these characteristics of the dataset, to enhance training accuracy and speed, we employed a transfer learning approach based on fine-tuning to train the proposed model. This strategy helps leverage existing knowledge under limited data conditions, enabling rapid and efficient model training to better adapt to the task of classifying images of defects in aluminum profiles.

In classifying images of defects on aluminum profiles, we chose to improve upon the ResNet network, which has outstanding classification performance. Specifically, ResNet50 was selected as the base network. This network comprises four major convolutional blocks, each containing three, four, six, and three small residual modules, respectively. Each residual module contains three layers, and when the input size is a 224 × 224 image, the output sizes of the convolutional blocks are 56 × 56, 28 × 28, 14 × 14, and 7 × 7, as detailed in Table 1, which describes the structure of the ResNet50 network.

#### 2.3.2. Dilated Convolution

Dilated convolution [33], compared to regular convolution, introduces an additional hyperparameter called dilation rate. The size of the dilation rate affects the receptive field of the convolution. The calculation formula for dilated convolution is as follows:(6)m=k+(k−1)×(v−1)
(7)$$o=\frac{i-2p-m}{l}+1$$

In the equation, *k* represents the size of the convolution kernel in the original convolution, *v* is the dilation rate, *m* is the effective size of the dilated convolution kernel, *p* is the padding during convolution, *l* is the stride, *i* is the input size, and *o* is the output size. Considering the potential issues of local information loss or weak correlation when selecting information over long distances, in this study, the original convolution kernel size for dilated convolution is set to three, as illustrated in Figure 4.

For the three convolutional blocks in the ResNet50 structure with output sizes of 56 × 56, 28 × 28, and 14 × 14, the output of each block is fed into an dilated convolutional layer to enhance the model’s feature extraction capability, as illustrated in Figure 5.

### 2.4. Construction of the AluDef-ClassNet Model

Based on Section 2.1, Section 2.2 and Section 2.3, we propose the AluDef-ClassNet network. The overall architecture of this network is illustrated in Figure 6.

## 3. Dataset

### 3.1. Composition of the Hardware

For the acquisition of the surface defect images of the aluminum profiles, We use a Basler ace acA1300-30gm camera from BASLER in Aachen, Germany, as illustrated in Figure 7. The Basler ace acA1300-30gm utilizes the Sony ICX445 CCD sensor, which is a commonly used imaging sensor technology in digital cameras and machine vision systems, providing high-quality images [34]. The CCD sensor is composed of an array of tiny photosensitive units (pixels). Each pixel contains a photodiode that converts incident light into electric charge. When light strikes the photodiode, it generates electrons proportional to the intensity of the light. These charges are then moved via charge transfer and ultimately read out and converted into digital signals, forming an image. The image data are transmitted to the connected computer through its GigE interface.

The Basler ace acA1300-30gm camera has a resolution of 1280 × 1024 pixels. It can capture images in real-time continuous mode or through trigger mode. In this study, we utilized real-time continuous image acquisition, continuously capturing images at fixed time intervals. This continuous mode is suitable for applications requiring continuous monitoring and detection, such as online quality inspection, surveillance, and real-time image processing.

Image acquisition primarily consists of a light source and an industrial camera. These complementary hardware components significantly influence imaging quality, which directly impacts the efficiency of machine vision-based defect recognition systems when running online. In our setup, we employed “cold light” sources using LED light-emitting diodes (LEDs). The role of the light source in the acquisition system is to enhance the detectable features of surface defects on aluminum profiles while suppressing irrelevant information in the surface images, thereby minimizing noise and environmental interference in the captured images to ensure image quality and stability.

During the acquisition process, the camera is vertically mounted at a fixed position at an appropriate distance from the object being inspected, allowing it to capture images of the entire object surface. We continuously monitor the image quality and visibility of defects in real-time, adjusting the position and angle of the light source as needed to ensure continuous high-quality image capture. The angles and directions of the light source are carefully adjusted to highlight the details and features of the aluminum profile surface to the fullest extent possible. The specific setup is illustrated in Figure 8.

### 3.2. Construction of the Dataset

After obtaining registration parameters, image data of the target are collected without changing the camera position. In this study, we utilized a sliding window of 224 × 224 pixels to extract the portions of the entire surface defect images captured by the CCD camera. The selection of complete defect images was performed to establish a standardized dataset of small aluminum profile surface defect images. Additionally, techniques such as image flipping, rotation, scaling, and translation were employed to augment the standardized image dataset, resulting in a total of 1380 standard sample images. These images were mainly categorized into seven classes, including non-conductive defects, scratches, transverse concave layer cracks, pinholes, indentations, and defect-free (as shown in Figure 9).

The dataset is divided into training, validation, and testing sets. The training set consists of 800 images, the validation set contains 180 images, and the testing set comprises 400 images, as detailed in Table 2.

## 4. Experimental Section

### 4.1. Setup Details

The experiments in this study were conducted on the Ubuntu 20.04.2 LTS operating system using the PyTorch deep learning framework and Python3.10 as the programming language. The CPU employed was the Intel Core i5-12490F, and the GPU utilized was the NVIDIA GeForce RTX3080.

During the training process, the model employed a transfer learning approach. Specifically, the model loaded the pre-trained parameters of the ResNet50 network on the ImageNet dataset, excluding the parameters of the fully connected layers used for classification in the ResNet50 network. For these unloaded fully connected layer parameters, the model utilized normal initialization for parameter initialization. This approach aims to leverage the universally learned features on the large-scale ImageNet dataset to expedite the model’s learning process for aluminum profile surface defect images. Simultaneously, normal initialization is applied to adapt to the specific requirements of the new task.

### 4.2. Algorithm Evaluation Indicators

When dealing with a six-classification problem, we can use the concepts of true positives (TPs), true negatives (TNs), false positives (FPs), and false negatives (FNs) to calculate various performance metrics. TPs and FPs represent true positives and false positives, respectively, while TNs and FNs represent true negatives and false negatives, respectively.

Accuracy (ACC) refers to the proportion of correctly classified samples by the model out of the total number of samples. It is calculated as the sum of all the TPs for each class divided by the total number of samples. The accuracy indicates the overall performance of the classification.

Recall measures the proportion of correctly predicted positives out of all the true positives. For each class, recall is calculated as the TPs for that class divided by the sum of the TPs and FNs for that class. Recall can be calculated for each class, and then the macro-average (simple average) or micro-average (global statistics) can be applied.

Precision gauges the proportion of true positives out of all the samples predicted as positive by the model. For each class, precision is calculated as TPs divided by the sum of the TPs and FPs for that class. Average precision (AP) is a metric mainly used for tasks like object detection. In multi-class problems, mean average precision (mAP) can be calculated using the area under the precision–recall curve (AUC-PR). For each class, we calculate the area under the precision–recall curve and then take the average.

The F1 score is the harmonic mean of the precision and recall. For each class, the *F*1 score can be calculated using Formula (8). The macro-average or micro-average of the *F*1 score provides an overall assessment of the model’s performance.
(8)$$F1=\frac{2\times Precision\times Recall}{Precision+Recall}$$

### 4.3. Analysis of Experimental Results

In this experiment, to obtain the optimal hyperparameters for the model, we explored different learning rates, as shown in Figure 10a. To achieve higher accuracy, we compared three optimization algorithms: AdamW [35], Adam [36], and SGD [37], as illustrated in Figure 10b.

We utilized the AdamW [35] optimization algorithm to update the model’s weights and minimize the loss function. Each batch consisted of 128 images. The weight decay was set to 0.001, the momentum factor to 0.9, the learning rate to 0.001, and the learning strategy was set to STEP. In GPU mode, a normalization factor was used to accelerate the training process, and the maximum number of training iterations was set to 100.

The accuracy curve, training loss curve, and validation loss curve of the AluDef-ClassNet model during the training process are shown in Figure 11. From the graph, it can be observed that with an increase in the number of iterations, the model’s accuracy on the validation set continuously improves, and the loss consistently decreases. The model reaches a satisfactory convergence, achieving 90% accuracy at 12 iterations.

The confusion matrix obtained by the model on the test set is presented in Table 3, serving as a crucial tool for evaluating prediction accuracy. The confusion matrix is a square matrix representing the relationship between actual and predicted categories. Rows of the matrix correspond to true categories, while columns represent predicted categories. Each element in the matrix indicates the number of instances classified into combinations of true and predicted categories.

The present study employed a progressive ablation experiment to further validate the effectiveness of the proposed model. Initially, a ResNet50 model with dilated convolutions was utilized as the baseline model. Subsequently, the Gaussian difference pyramid structure, self-attention mechanism, and dilated convolutions were progressively added, and ablation experiments were conducted accordingly. The results are summarized in Table 4.

Table 2’s second row demonstrates that the proposed Gaussian difference pyramid module effectively enhances the model’s performance. This is attributed to its capability to capture multiscale image features, thereby improving the model’s ability to discern subtle and inconspicuous defects. The results in the third row indicate that embedding the self-attention mechanism during feature extraction significantly boosts the defect recognition performance. Additionally, the fourth row reveals a slight improvement in accuracy by incorporating a dilated convolution within the residual block. The integration of the Gaussian difference pyramid, self-attention mechanism, and enhanced residual networks has collectively resulted in performance enhancement.

We compared the accuracy of our model with other models, as shown in Figure 12. After 100 iterations, AlexNet [13], VGG [14], ResNet-18 [15], ResNet-50 [38], ResNet-101 [39], YOLOV5 [40], and AluDef-ClassNet achieved classification accuracies of 87.5%, 86.6%, 91.8%, 94.3%, 93.7% 94.1%, and 97.6%, respectively. Therefore, this model can effectively identify and classify different types of defects on the aluminum surface, demonstrating high performance and reliability.

To investigate the impact of different modules on network performance, five evaluation metrics were introduced in this study: feature memory, computational capability (number of floating-point operations per second (FLOPs)), recall rate, *F*1 score, and mAP accuracy. AluDef-ClassNet was compared with six popular classical networks. Table 5 presents detailed comparison results.

The results above fully demonstrate that this model improves the effectiveness and robustness of defect detection. The Gaussian difference pyramid and self-attention mechanism contribute to a better understanding of the correlation in different regions of the image. The improved residual network can capture broader contextual information, aiding in handling global features in the image, and enhance the accuracy of defect detection.

## 5. Conclusions

This study focuses on the classification of surface defects in aluminum profiles, proposing the AluDef-ClassNet model. Through using the Gaussian difference pyramid, the self-attention mechanism, and the improved residual network, the model effectively addresses challenges such as multiscale variations, lighting changes, occlusions, and noise encountered in aluminum profile surface defect detection. The experimental results demonstrate that the model significantly surpasses classical models in various performance metrics, exhibiting high classification accuracy and robustness. However, it is noted that insufficient sample quantities may impact the detection performance for specific defect categories, despite the application of data augmentation. Therefore, while AluDef-ClassNet brings a notable breakthrough to surface defect detection in aluminum profiles, attention to sample distribution and data scale effects on model performance remains essential in practical applications. Future research could focus on further optimizing model structures, exploring larger datasets, and addressing extreme sample distributions to enhance the model’s robustness and generalization performance. This study provides valuable empirical insights for the application of deep learning in aluminum profile surface defect detection and outlines directions for future research.

## Figures and Tables

**Figure 1 sensors-24-02914-f001:**
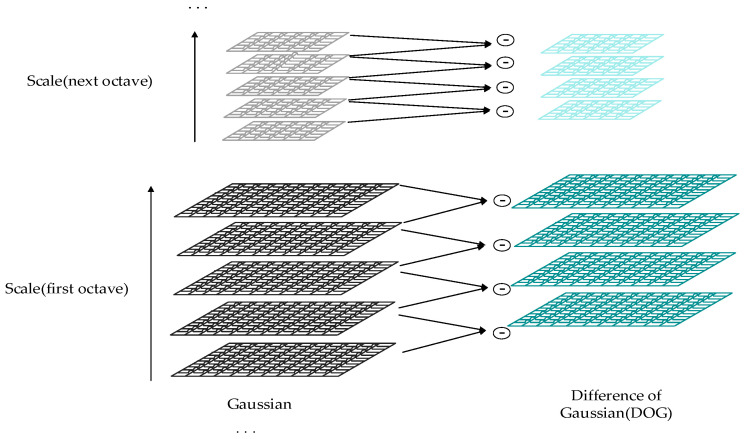
Structure diagram of the Gaussian difference pyramid.

**Figure 2 sensors-24-02914-f002:**
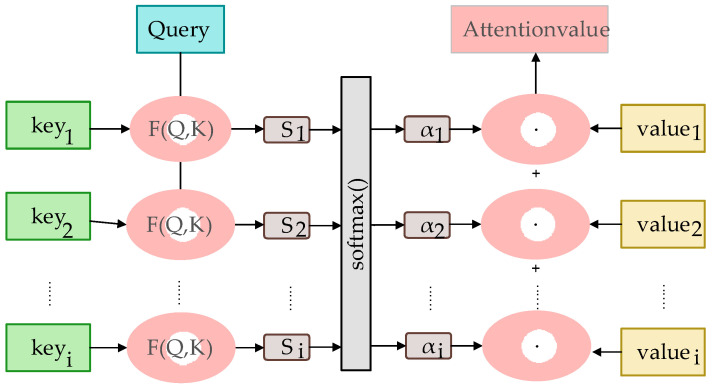
Flowchart of the attention mechanism.

**Figure 3 sensors-24-02914-f003:**
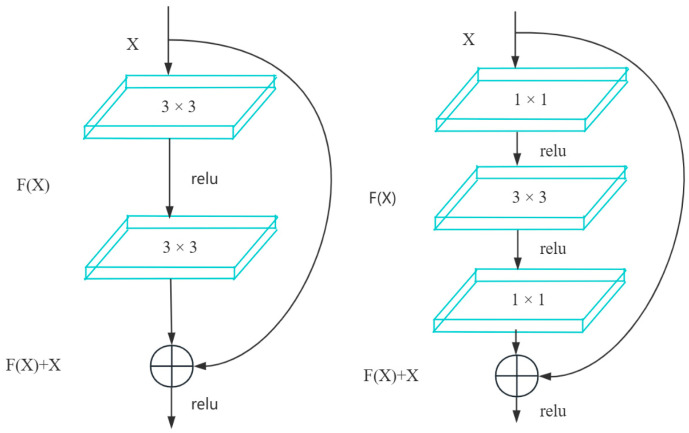
Residual block.

**Figure 4 sensors-24-02914-f004:**
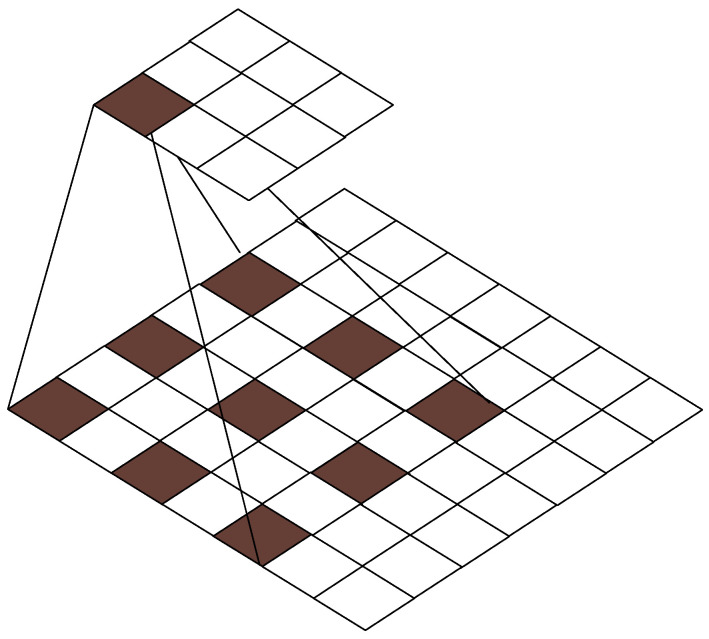
Dilated convolution.

**Figure 5 sensors-24-02914-f005:**
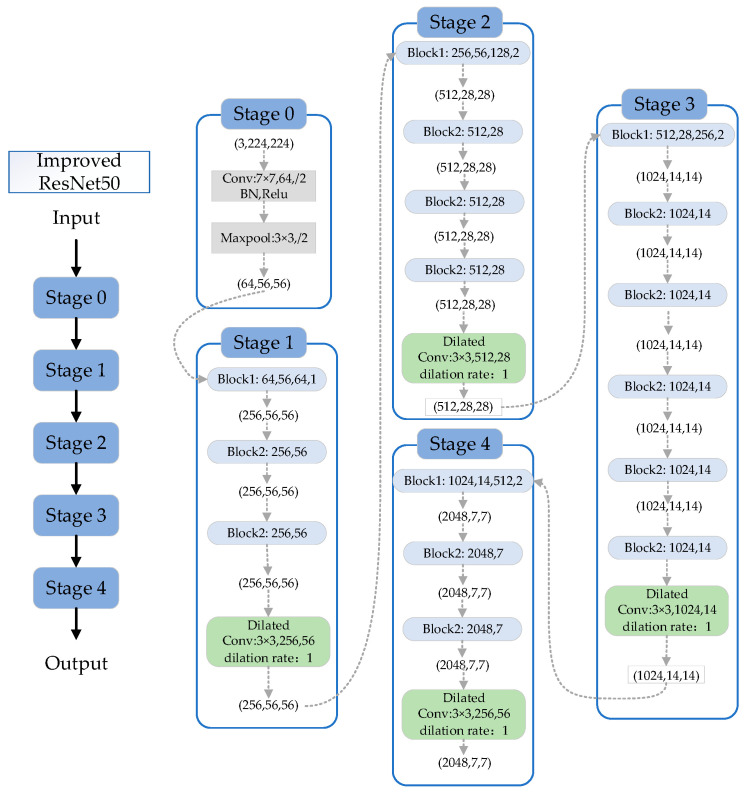
The enhanced ResNet50 network architecture.

**Figure 6 sensors-24-02914-f006:**
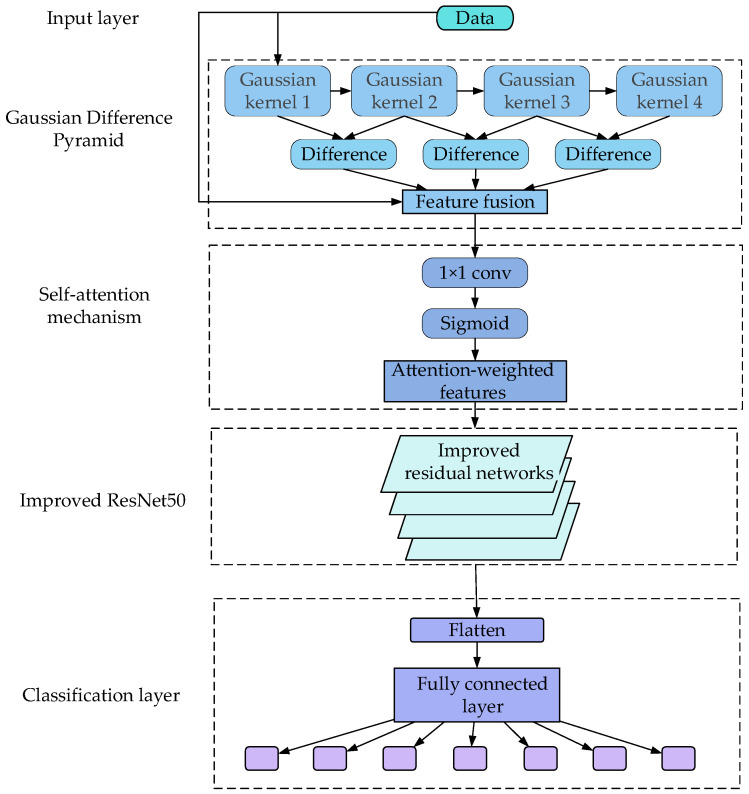
Flowchart of the AluDef-ClassNet network.

**Figure 7 sensors-24-02914-f007:**
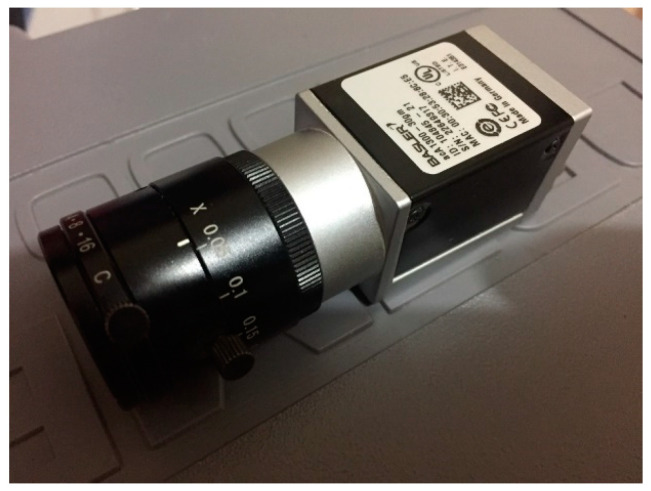
Exterior diagram of the CCD camera.

**Figure 8 sensors-24-02914-f008:**
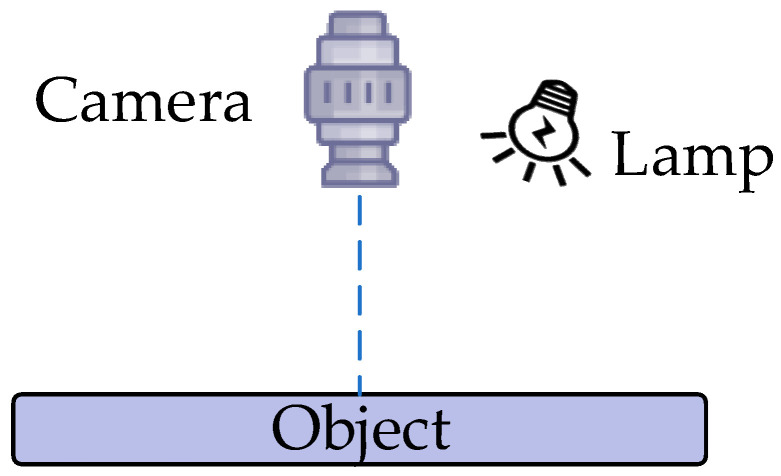
Hardware placement position.

**Figure 9 sensors-24-02914-f009:**
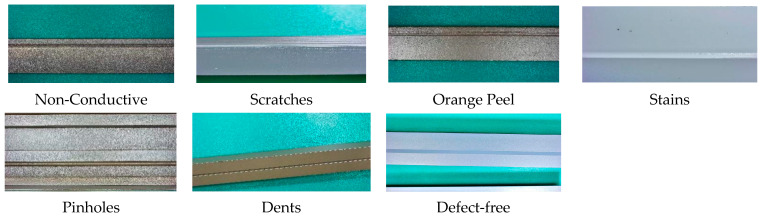
Defect types.

**Figure 10 sensors-24-02914-f010:**
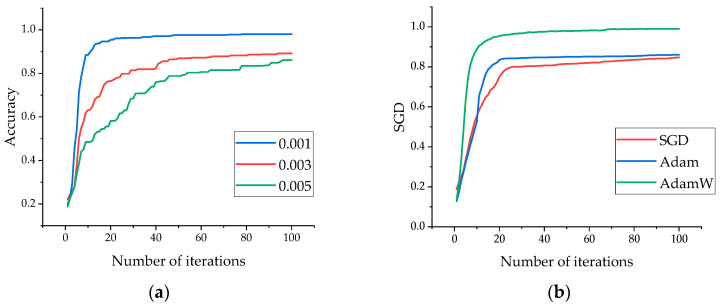
Comparison of different learning rates and optimization methods. (**a**) Comparison of different learning rates. (**b**) Comparison of different optimization methods.

**Figure 11 sensors-24-02914-f011:**
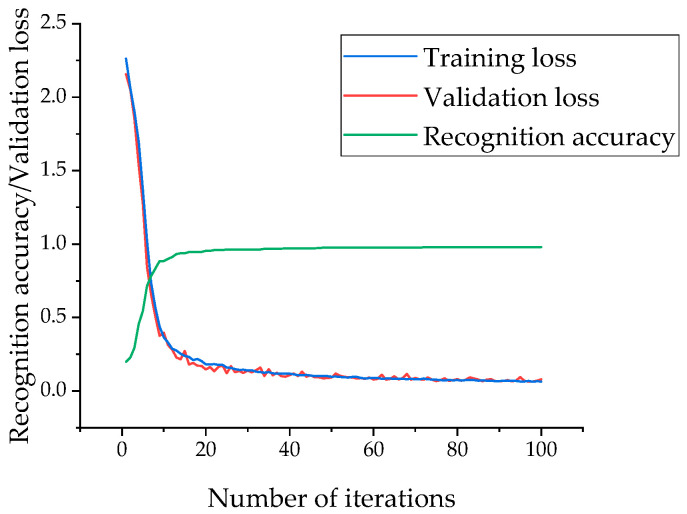
Defect image recognition accuracy and loss curves.

**Figure 12 sensors-24-02914-f012:**
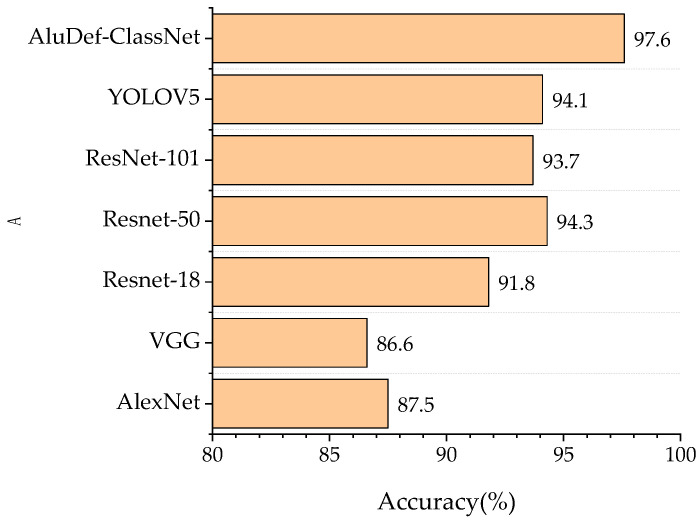
Comparison of the accuracy of each model.

**Table 1 sensors-24-02914-t001:** The structure of the ResNet50 network.

Model	Output Size	Specific Layer
Convolutional block 1	56 × 56	$\left(\begin{array}{cc} 1\times 1, & 64\\ 3\times 3, & 64\\ 1\times 1, & 256 \end{array}\right)\times 3$
Convolutional block 2	28 × 28	$\left(\begin{array}{cc} 1\times 1, & 128\\ 3\times 3, & 128\\ 1\times 1, & 512 \end{array}\right)\times 4$
Convolutional block 3	14 × 14	$\left(\begin{array}{cc} 1\times 1, & 256\\ 3\times 3, & 256\\ 1\times 1, & 1024 \end{array}\right)\times 6$
Convolutional block 4	7 × 7	$\left(\begin{array}{cc} 1\times 1, & 512\\ 3\times 3, & 512\\ 1\times 1, & 2048 \end{array}\right)\times 3$

**Table 2 sensors-24-02914-t002:** Data distribution.

Data Distribution	Non-Conductive	Scratches	Orange Peel	Stains	Pinholes	Dents	Defect-Free	Total
Training set	40	136	90	209	248	77	100	900
Validation set	10	30	20	50	50	20	20	200
Test set	25	77	54	96	105	43	50	450

**Table 3 sensors-24-02914-t003:** Confusion matrix.

	Predicted to Be Non-Conductive	Predicted to Be Scratches	Predicted to Be Orange Peel	Predicted to Be Stains	Predicted to Be Pinholes	Predicted to Be Dents	Predicted to Be Defect-Free
Non-conductive	24	1	0	0	0	0	0
Scratches	1	76	0	0	0	0	1
Orange peel	0	0	52	1	1	0	1
Stains	0	1	0	94	0	1	0
Pinholes	1	1	0	0	103	0	1
Dents	0	0	0	1	0	42	0
Defect-free	0	0	1	0	0	0	49

**Table 4 sensors-24-02914-t004:** Ablation experiment.

Method	*F*1/%	*Recall*/%	*Precision*/%
Baseline	92.2	90.2	94.3
Baseline + Gaussian difference pyramid	86.27	83.76	94.79
Baseline + Gaussian difference pyramid + self-attention	88.06	86.07	95.93
Baseline + dilated convolution	87.26	85.97	94.43
AluDef-ClassNet	89.26	88.69	97.6

**Table 5 sensors-24-02914-t005:** Comparison of the comprehensive performance of different networks.

Model	Feature Memory	Number of Floating-Point Operations (FLOPs)	Recall	F1	mAP
AlexNet	96 MB	0.72 GFLOPs	82.3%	84.8%	87.2%
VGG	95 MB	15.5 GFLOPs	81.9%	84.2%	85.2%
Resnet-18	11 MB	1.8 GFLOPs	87.6%	89.7%	92.1%
Resnet-50	155 MB	10 GFLOPs	90.2%	92.2%	95.2%
ResNet-101	305 MB	19.6 GFLOPs	89.8%	91.7%	93.6%
YOLOV5	217 MB	11.2 GFLOPs	90.5%	92.2%	95.7%
AluDef-ClassNet	235 MB	15.2 GFLOPs	94.2%	95.4%	98.6%

## Data Availability

Data are contained within the article.

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
