# Peer review of "Surface Defect Detection of Aluminum Profiles Based on Multiscale and Self-Attention Mechanisms"

_sensors, 2024, doi:10.3390/s24092914_

Round 1

Reviewer 1 Report

Comments and Suggestions for Authors

In order to accurately identify surface defects on aluminum, this article proposes the AluDef-ClassNet model, which combines Gaussian difference pyramid, self attention mechanism, and improved residual network. The article has achieved certain results, but there are still the following issues:

1. Some images in the text have smaller fonts, such as Figures 5, 9, 10, etc. In addition, the font in the image is not consistent.

2. Will lighting angle affect image details? How is it specifically set up?

3. Is there any phenomenon of imbalanced data after data augmentation? Suggest adding an enhanced data description.

4. This article should add a set of ablation experiments to demonstrate the effectiveness of Gaussian difference pyramid, self attention mechanism, and improved residual network, respectively.

5. Is there a visual result graph available?

6. The article should be compared with the most advanced deep learning networks currently available.

7. The article lacks sufficient description of innovation and seems to be a simple addition to existing technologies. It is recommended to carefully polish the second section.

Comments on the Quality of English Language

Minor editing of English language required

Reviewer 2 Report

Comments and Suggestions for Authors

The present paper introduces a methodology for automatically detecting defects in aluminum profiles using Deep Learning architectures. However, there are some points that need clarification.

  • While classification is performed among different types of defects, it's important to address how the methodology handles defect-free scenarios.

  • The proposed methodology employs a modified version of ResNet50, designed to differentiate among a large number of classes. However, using such a network for the classification of only six classes may not be efficient from a computational burden standpoint. This complexity is compounded by the use of pyramids, feature fusion, and attention mechanisms. It appears that the proposed solution may be overly complex for the classification task at hand. It would be beneficial to provide results in terms of computational burden and compare the methodology with simpler approaches such as few-layer CNNs.

  • How practical is this methodology for industrial environments? Is it feasible to embed the machine learning model within low-cost microcontrollers?

Comments on the Quality of English Language

Seems fine. Just proof-reading required.

Round 2

Reviewer 2 Report

Comments and Suggestions for Authors

The authors addressed all the queries.